# Improving Model Alignment Through Collective Intelligence of Open-Source LLMS

## Abstract

Building helpful and harmless large language models (LLMs) requires effective model alignment approach based on human instructions and feedback, which necessitates high-quality human-labeled data. Constructing such datasets is often expensive and hard to scale, and may face potential limitations on diversity and generalization. To address these challenges, we introduce Mixture of Agents Alignment (MoAA), that leverages the collective strengths of various language models to provide high-quality data for model alignment. By employing MoAA, we enhance both supervised fine-tuning and preference optimization, leading to improved performance compared to using a single model alone (e.g. using GPT-4o alone). Evaluation results show that our approach can improve win rate of LLaMA-3.1-8B-Instruct from 19.5 to 48.3 on Arena-Hard and from 22.33 to 57.23 on AlpacaEval2, highlighting a promising direction for model alignment through this new scalable and diverse synthetic data recipe.[1]

## 1 Introduction

Model alignment is a crucial stage of training large language models (LLMs) towards their safe and helpful deployment (Ouyang et al., 2022a; Bai et al., 2022). A well-established model alignment protocol includes supervised finetuning (SFT) (Zhang et al., 2023) and reinforcement learning with human feedback (RLHF) (Casper et al., 2023). During the SFT stage, models imitate the human-level responses by learning from an instruction dataset; hence, the data quality often determines the finetuned model's instruction following capability. Following the SFT stage, RLHF further enhances the model alignment by constructing a reward model that emulates human preferences, based on which policy optimization is conducted to maximize the reward objective (Ouyang et al., 2022a). Direct preference optimization (DPO) further simplifies the RLHF strategy by directly optimizing LLMs on the preference data and learning an implicit reward function, which is proved to be effective on model alignment (Rafailov et al., 2023). The quality for both instruction and preference data determines the performance of model alignment.

To alleviate the high cost of human-crafted datasets (Köpf et al., 2023; Zhou et al., 2023; Longpre et al., 2023), synthetic data (Ding et al., 2023; Taori et al., 2023; Wang et al., 2023c) can be created by automating the response collection process via stronger LLMs such as GPT-4 (OpenAI, 2023a). However, the quality and potential biases from a single strong model may deteriorate the alignment performance (Shumailov et al., 2024). Another challenge lies on the black-box nature of proprietary LLMs, raising research reproducibility concerns (Chen et al., 2023). Fortunately, an increasing number of open-source LLMs have been released (Dubey et al., 2024; Bai et al., 2023b; Xu et al., 2023a; Jiang et al., 2024; Team et al., 2024), with expertise in various aspects and

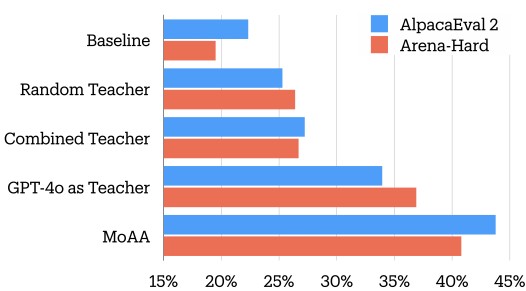

Figure 1: SFT results using different models to generate synthetic data. Baseline is the original LLaMA-3.1-8B-Instruct model.

---

[1]We will release the data and code used in this work.

tasks. It is intriguing to leverage these open-source models jointly for model alignment due to the their intrinsic diversity. Taking SFT as an example, a naive approach is to align a base model with the outputs from a group of open-sourced models (teachers). One method is to combine data generated by five different models into one synthetic tuning set (*Combined Teacher*), or randomly sample a model to generate for each instruction in a tuning set (*Random Teacher*). However, these methods often do not yield satisfactory results and may still be worse than using a single more capable proprietary model to generate synthetic data, as shown in Figure 1.

Mixture of Agents (MoA) offers new opportunities in leveraging collective intelligence of open-source LLMs (Wang et al., 2024c). For example, MoA built solely on open-sourced models outperforms state-of-the-art proprietary models on chat-based benchmarks such as AlpacaEval (Dubois et al., 2024). Despite these promising results, the integration of MoA into the model alignment process to further leverage benefits of the open-source LLMs remains under-explored.

In this work, we propose Mixture of Agents Alignment (MoAA), an effective alignment recipe that leverages the collective intelligence of multiple open-source LLMs to generate high-quality synthetic data. Our approach consists of a two-stage training scheme, which we refer as MoAA-SFT and MoAA-DPO. In the first stage, we employ a diverse ensemble of open-source models to generate synthetic SFT data, and then conduct SFT. This diverse and high-quality data significantly enhances the performance of the fine-tuned model compared to data generated from a single model or less diverse datasets. The high quality of MoA responses brings promises for model alignment, as can be seen from the SFT result of MoAA in Figure 1. Following SFT, we apply DPO to further refine the model's alignment with human preferences, improving its ability to generate helpful and harmless responses. Specifically, we sample multiple responses from the SFT model and use another combination of MoA as reward model to decide the chosen / rejected responses.

Our evaluation on benchmarks AlpacaEval2, Arena-Hard, MT-Bench shows significant improvements, highlighting the effectiveness of MoAA. Notably, we observe a substantial increase in the win rates of both LLaMA-3.1-8B-Instruct and Gemma-2-9B-It, sometimes even matching the Length-Controlled (LC) win rate of the MoA model used to generate the data, on AlpacaEval2.

We summarize our contributions as follows:

(1) **SFT Data Generation Pipeline**: We proposed to generate high-quality synthetic SFT data using the MoA approach, which leverages the collective strengths of multiple open-source LLMs.
(2) **DPO Preference Annotation Pipeline**: We proposed an adapted MoA setup to annotate preference data for effective DPO, eliminating the need for training an additional reward model.
(3) **Extensive Evaluation**: We conducted comprehensive evaluations on multiple benchmarks, demonstrating significant improvements in response quality.
(4) **Data and Model Release**: We will release our instruction data, preference data, and the code used to generate them. We hope this will facilitate further research and development in the field of model alignment.

## 2 RELATED WORK

**Model Alignment**. LLMs trained on large datasets acquire surprising capabilities (Brown et al., 2020; OpenAI, 2023a; Touvron et al., 2023a;b; Chowdhery et al., 2022; Anil et al., 2023; Kaplan et al., 2020; Brown et al., 2020; OpenAI, 2023b). To leverage these capabilities to real applications, pre-trained LLMs usually needs to be further fine-tuned on instruction data (Köpf et al., 2023; Zhou et al., 2023; Longpre et al., 2023; Ding et al., 2023; Taori et al., 2023; Wang et al., 2023c). Such alignment process can be roughly categorized into supervised fine-tuning (SFT, Zhang et al. 2023) and reinforcement learning from human feedback (RLHF, Ouyang et al. 2022b). SFT directly training on the instruction data with cross-entropy loss, is one of the effective way to gain the ability to interact with humans. Using SFT as a precedent step, RLHF (Ouyang et al., 2022a; Bai et al., 2022) aligns further with human preferences and societal well-being (Russell & Norvig, 2020; Russell, 2022). Popular RLHF approaches include proximal policy optimization (PPO) (Schulman et al., 2017), direct preference optimization (DPO) (Rafailov et al., 2023), KTO (Ethayarajh et al., 2024), $\psi$PO (Gheshlaghi Azar et al., 2024), etc.

**Model Ensemble**. As open-source and proprietary large language models become more accessible, it is intriguing to leverage the collective intelligence of existing models. Model merging, ensemble

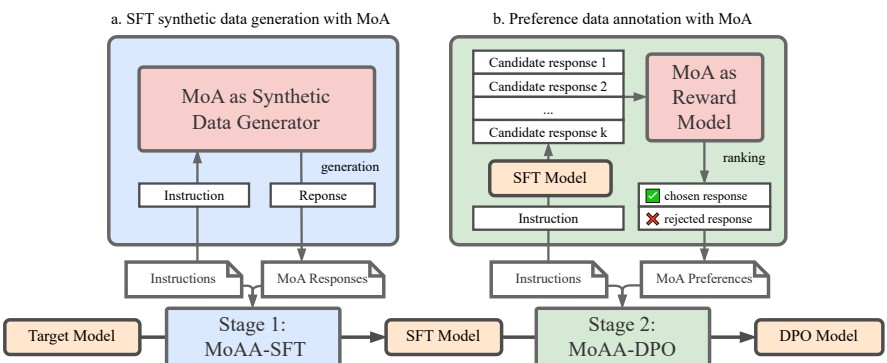

Figure 2: Two-stage Mixture of Agents Alignment to enhance the target model performance.

and cooperation, e.g. multi-agent (Guo et al., 2024), are several promising directions of collaborative strategies of multiple LLMs (Lu et al., 2024). In particular, one simple model ensemble method is repeated sampling, which proves to be helpful in commonsense reasoning (Wang et al., 2023b) and coding tasks (Brown et al., 2024). The model ensemble approach has gained more popularity recently due to the release of GPT4 o1 (OpenAI, 2024), where scaling up the inference compute (Snell et al., 2024) and performing effective sampling/search approach boost model performance on high-complexity tasks such as science, coding and mathematics. On the other hand, Mixture of Agents (MoA) (Wang et al., 2024c) leverages the diversity and capabilities of open-source models and proposes a layered proposer-aggregator architecture to iteratively refine the model ensemble outputs. MoA built on open-source LLMs outperforms state-of-the-art proprietary LLMs in chat-related benchmarks, offering new opportunities of augmenting open-sourced LLMs.

## 3 MIXTURE OF AGENTS ALIGNMENT METHODOLOGY

In this section, we detail our two-stage Mixture of Agents Alignment methodology designed to enhance the target model's performance, as shown in Figure 2. In the first stage, we employ MoA (Wang et al., 2024c) to produce high-quality synthetic data for supervised fine-tuning. The second stage combines multiple LLMs as a reward model to provide preference annotations.

### 3.1 STAGE 1: SUPERVISED FINE-TUNING VIA MOAA

We begin by introducing the MoA approach, specifically how LLMs can collaborate to generate high-quality responses. Then we will demonstrate the enhanced instruction tuning with MoA-generated synthetic data.

### 3.1.1 MIXTURE OF AGENTS

LLMs have demonstrated a remarkable capacity for collaboration, producing higher-quality responses when they can reference other models' outputs in a structured manner. To maximize the benefits of such multi-model collaboration, it is crucial to design a framework that effectively characterizes and fully utilizes the unique expertise of different LLMs. The Mixture of Agents strategy exemplifies this approach by categorizing LLMs into distinct roles:

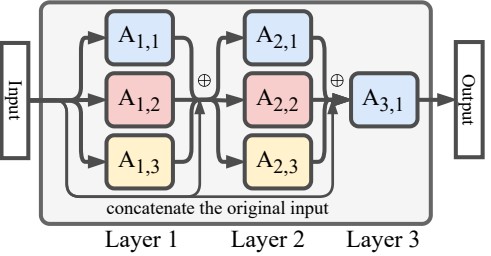

Figure 3: The architecture of Mixture-of-Agents (Wang et al., 2024b). This example showcases 3 MoA layers where the first layer has three proposers, the second layer has three aggregators that also serve as proposers in the next layer, and the last layer has one aggregator.

**Proposers** excel at generating useful reference responses for use by other models. While a good proposer may not necessarily produce responses with high scores by itself, it offers more context and diverse perspectives, contributing to better final responses when used by an aggregator.

**Aggregators** are models proficient in synthesizing responses from other models into a single, high-quality output. An effective aggregator should maintain or enhance output quality even when integrating inputs that are of lesser quality than its own.

Formally, it has $l$ layers and each layer-$i$ consists of $n$ LLMs, denoted by $A_{i,1}, A_{i,2}, ..., A_{i,n}$. Each LLM $A_{i,j}$ processes an input text and generates its continuation. Formally, given an input prompt $x_0$, the output $y_{i,j}$ of $i$-th MoA layer for LLM $A_{i,j}$ can be expressed as follows:

$$y_{i,j} = A_{i,j}\left([\text{context}] + \oplus_{k=1}^{n} y_{i-1,k} + x_0\right), \quad y_{0,j} = A_{1,j}\left([\text{context}] + x_0\right) \tag{1}$$

where $+$ here means concatenation of texts; [context] represents optional additional context; $\oplus$ means application of the Aggregate-and-Synthesize prompt shown in Table 18 to model outputs.

### 3.1.2 SYNTHETIC DATA GENERATION FROM MoA

We leverage MoA to generate high-quality synthetic data for SFT. Given an instruction $q_0$ from an instruction-tuning set, we process it through the MoA framework. We abstract this process defined by Equation 1 as $\mathcal{M}_{\text{SynGen}}(\text{instruction}, \#\text{ layers}, [\text{context}])$. The synthetic response is obtained via:

$$y_l = \mathcal{M}_{\text{SynGen}}(q_0, l, null) \tag{2}$$

where $y_l$, the output from the final layer, is the synthetic response, incorporating insights from all proposer and aggregator models. In practice, we employ a two-layer MoA approach to expedite the process, as it is sufficient to generate high-quality synthetic data.

**Multi-Turn Instructions** For multi-turn instructions, we synthesize responses for each query sequentially. Formally, given the current instruction prompt $q_k$ and previous instructions with their MoA synthesized responses, the MoA synthesized data for the current turn can be expressed as:

$$y_l^k = \mathcal{M}_{\text{SynGen}}(q_k, \; l, \; q_1 + y_l^1 + q_2 + y_l^2 + \cdots + y_l^{k-1}) \tag{3}$$

where we concatenate previous turns using $+$ and $k$ represent which turn. Note that there are other ways to design the architecture, e.g., we can decide whether to put the previous turns' context before or after the MoA prompt. We leave a more exhaustive search of optimal structure to future work. Note that some of the multi-turn data may suffer from the problem of discontinuity. That is, the next query may depend on the previous responses. In practice, we do not observe this to be too much of a problem in the dataset we used, but we think in the future, a more sophisticated and granular way of generating multi-turn data can be deployed.

### 3.2 STAGE 2: PREFERENCE ALIGNMENT FROM MoAA

The second stage of our Mixture of Agents Alignment process adapts MoA as a reward model for labeling the preference alignment dataset. In this section, we will (1) give a brief overview of DPO and its use in model alignment; (2) detail our approach to reward modeling; (3) introduce an additional criteria filtering step that further enhances performance.

### 3.2.1 DIRECT PREFERENCE OPTIMIZATION (DPO)

DPO (Rafailov et al., 2023) is one of the most commonly used offline preference optimization methods. Instead of learning a reward model and then optimizing it via reinforcement learning like the conventional RLHF methods, DPO reparameterizes the reward function that enables the extraction of its optimal policy in a closed form:

$$r(x, y) = \beta \log \frac{\pi_\theta(y \mid x)}{\pi_{\text{ref}}(y \mid x)} + \beta \log Z(x) \tag{4}$$

where $\beta$ is a hyperparameter, $\pi_\theta$ is the policy model and $\pi_{\text{ref}}$ is the reference policy model. By incorporating this into Bradley-Terry model, we can get the DPO objective to be:

$$\mathcal{L}_{\text{DPO}}(\pi_\theta; \pi_{\text{ref}}) = -\mathbb{E}_{(x, y_w, y_l) \sim \mathcal{D}} \left[ \log \sigma \left( \beta \log \frac{\pi_\theta(y_w | x)}{\pi_{\text{ref}}(y_w | x)} - \beta \log \frac{\pi_\theta(y_l | x)}{\pi_{\text{ref}}(y_l | x)} \right) \right] \tag{5}$$

where $x$ is the instruction, $y_w$ is the winning response and $y_l$ is the losing response from preference data $\mathcal{D}$. To construct those preference pairs for DPO, during our preference data annotation process, we first sample completions $y_i \sim \pi_{\text{ref}}(\cdot \mid x)$ from our reference model which is the SFT model given instruction $x$. Then we use our MoA as a reward model to pick the highest-scoring response as $y_w$ and lowest-scoring response as $y_l$, as detailed in the next section.

### 3.2.2   MoA as a Reward Model

We employ the MoA architecture as a reward model for preference alignment, to address limitations of traditional single-model approaches. Our method leverages original open-source LLMs without specific reward modeling training, which are combined in the MoA architecture to harness collective intelligence. The structure mirrors that of the data synthesis stage, featuring LLMs as both proposers and aggregators:

**Proposers** generate balanced and comprehensive assessments of response quality. We design a specific prompt different from the one in SFT stage, as detailed in Table 20.

**Aggregators** synthesize the evaluations provided by proposers to render a final judgment, complete with clear reasoning. The specific prompt used for aggregators can be found in Table 21. Our evaluation methodology employs a pairwise comparison approach, as Large Language Models (LLMs) have demonstrated superior performance in pairwise evaluations (Qin et al., 2023). To mitigate position bias (Wang et al., 2023a), each example undergoes dual evaluation, with the order of responses reversed in the second round. This approach ensures a more robust and unbiased assessment.

### 3.2.3   Criteria Filtering

Building upon previous work of Wang et al. (2024a), we incorporate a criteria filtering step to customize the evaluation for each query-response pair. Our approach differs in that we do not train models specifically for filtering. Instead, we prompt them to dynamically select relevant criteria:

1. We first prompt the model to analyze the user query and candidate responses, selecting the most relevant evaluation criteria from a predefined list in Table 19.

2. These selected criteria are then incorporated into the prompts for both proposer (Table 20) and aggregator (Table 21) models described in Section 3.2.2.

The rationale behind this filtering process is that different query types require distinct evaluation focuses. For example: (a) For potentially harmful queries (e.g., "how to build a bomb"), criteria like "Instruction adherence" or "Helpfulness" become inappropriate. In such cases, "Safety" would likely be prioritized; (b) Factual queries might weigh "Accuracy" more heavily; (c) Complex problem-solving tasks could emphasize "Depth" and "Robustness".

This dynamic selection ensures that the evaluation process adapts to the specific considerations of each query-response pair, leading to more nuanced and appropriate assessments.

The effectiveness of our criteria filtering approach is demonstrated in Table 22, showing improved performance on RewardBench (Lambert et al., 2024), particularly in Safety and Reasoning categories. This dynamic criteria selection is more robust and adaptive, capable of contextually relevant evaluations across diverse query types. It is used by default for subsequent evaluations.

## 4   Evaluation

We present our findings through a comprehensive evaluation in this section.

1. We achieve significant improvements on AlpacaEval 2 (Dubois et al., 2024), MT-Bench (Zheng et al., 2023), and Arena Hard (Li et al., 2024) benchmarks, with contributions from both SFT and DPO stages.

2. Extensive ablations are conducted to demonstrate the efficacy of our approach and provide insights into the relative contribution of each stage.

## 4.1 SETUP

**Models** We constructed MoA for data synthesis and response evaluation using various open-source LLMs and fine-tuned open-source models to enhance their capabilities. Our approach is not limited to open-source models and can be easily extended to closed-source models or a combination of both. In the first stage (supervised fine-tuning, SFT), we utilize a two-layer MoA architecture that uses WizardLM-8x22B (Xu et al., 2023b), Qwen2-72B-Instruct (Yang et al., 2024), Gemma-2-27B-it (Team et al., 2024), LLaMA-3.1-70B-Instruct (Dubey et al., 2024) as proposers and Qwen1.5-110B-Chat (Bai et al., 2023a) as aggregator. For the second stage (Direct preference optimization, DPO), a different two-layer mixture is used. Proposers include Gemma-2-27B-it, LLaMA-3.1-70B-Instruct, Qwen2-72B-Instruct and we use Qwen2-72B-Instruct again as the aggregator. We empirically search for an optimal architecture (selection of models in each layer) detailed in appendix B. A smarter discrete optimization method can be used to further increase performance but is out of the scope of this work. For open-source models, all inferences were run through Together Inference Endpoint.[2]

We apply our approach to two off-the-shelf instruction-tuned models: LLaMA-3.1-8B-Instruct, and Gemma-2-9B-it. We pick these open-source models to demonstrate that our approach can generalize to the state-of-the-art models.

**Training setups** During SFT in the first stage, we use a learning rate of $8.0e$-6 and batch size of 128 for both llama and gemma models. For LLaMA-3.1-8B-Instruct, we train for 6 epochs, and for Gemma-2-9B-it we train for 5 epochs. Packing is used as we found that it offers better improvement. In terms of the instruction set, we mainly utilize Ultrafeedback (Cui et al., 2023) for both models. We also add a 5,000 subset of Ultrachat-200k (Ding et al., 2023) to improve multi-turn capability. We limited the UC subset to 5,000 samples to prioritize efficiency while maintaining the desired performance improvements. We later present an ablation study on different mixtures of instruction tuning sets which can provide insights into why we choose this setup.

For DPO in the second stage, we use a learning rate of $8.0e$-7 for the llama model and a learning rate of $3.0e$-7 for the gemma model. We use a $\beta$ value of $0.01$ for both models. More details about hyper-parameters can be found in Appendix A. We subsampled 6,000 instructions from Ultrafeedback as the preference optimization set for DPO. To mitigate the distribution shift between SFT models and the preference alignment process, we generate the preference responses using the SFT models tuned by our MoA methods following the approach proposed by Meng et al. (2024). For each instruction, we generate 5 responses using the SFT model with a sampling temperature of 0.8. We then use our MoA reward model to score the 5 responses, selecting the highest-scoring one as the chosen response and the lowest-scoring one as the rejected response. Since our MoA reward model does pairwise evaluation, we compare all possible pairs out of 5 responses to acquire a ranking among those 5. This resulted in a total of 10 comparisons for each instruction.

**Benchmarks** Our evaluation primarily focuses on two leading benchmarks for assessing LLM alignment with human preferences: AlpacaEval 2 (Dubois et al., 2024) and Arena-Hard (Li et al., 2024). Both benchmarks employ a direct comparison methodology, pitting each model's response against that of GPT-4. Specifically, AlpacaEval 2 utilizes `gpt-4-1106-preview`, while Arena-Hard employs `GPT-4-0314`. A GPT-4-based evaluator then determines the preferred response, ensuring a consistent and high-quality assessment.

AlpacaEval 2 comprises 805 instructions that closely mirror real-world use cases. It implements length-controlled (LC) win rates to effectively neutralize length bias, a common confounding factor in language model evaluation. This metric has demonstrated remarkable alignment with human preferences, achieving a Spearman correlation of 0.98 with actual human evaluations (Dubois et al., 2024). Arena-Hard-Auto targets the evaluation of models on 500 challenging and demanding instructions submitted by real users in Chatbot Arena, thus maintaining a strong correlation with human preferences in complex scenarios.

To comprehensively assess multi-turn capabilities and performance across diverse domains, we additionally employ MT-Bench (Zheng et al., 2023). Unlike the comparative approach of AlpacaEval 2 and Arena-Hard-Auto, MT-Bench utilizes GPT-4 to grade model responses directly, without com-

---

[2]https://api.together.ai/playground/chat

Table 1: Model performances after applying our MoA alignment approach. We demonstrate MoAA-SFT and MoAA-DPO performances for both Llama and Gemma models. *MoA-Data-Generator* row showcases the performance of MoA directly on the benchmarks.

| Method | Size | AlpacaEval 2 (LC) | MT-Bench | Arena-Hard |
|---|---|---|---|---|
| Mistral-7B-instruct-v0.3 | 7B | 19.88 | 7.59 | 16.3 |
| Llama-3.1-8B-Instruct | 8B | 22.33 | 8.01 | 19.5 |
| Qwen2.5-7B-Instruct | 7B | 19.91 | 8.22 | 24.6 |
| Gemma-2-9B-it | 9B | 47.43 | 8.48 | 42.0 |
| Gemma-2-27B-it | 27B | **52.28** | 8.86 | 54.4 |
| Llama3.1-70B-Instruct | 70B | 37.26 | 8.99 | 55.2 |
| Qwen2-72B-Instruct | 72B | 38.10 | 8.88 | 45.0 |
| Qwen1.5-110B-Instruct | 110B | 43.90 | 8.96 | 56.4 |
| WizardLM-8x22B | 8x22B | 51.30 | 8.78 | **71.3** |
| Llama3.1-405b-Instruct | 405B | 40.19 | **9.18** | 61.5 |
| | | | | |
| Llama-3.1-8B-Instruct-MoAA-SFT | 8B | 43.77 | 8.33 | 40.8 |
| Llama-3.1-8B-Instruct-MoAA-DPO | 8B | 57.23 | 8.58 | 48.3 |
| Gemma-2-9B-it-MoAA-SFT | 9B | 53.79 | 8.65 | 47.6 |
| Gemma-2-9B-it-MoAA-DPO | 9B | **63.75** | **8.91** | **55.6** |
| | | | | |
| *MoA-Data-Generator (Reference)* | - | *62.50* | *9.17* | *75.9* |

parison to human-generated answers. This benchmark encompasses multi-turn instructions spanning eight distinct domains, including reasoning, writing, and knowledge. By incorporating MT-Bench, we gain deeper insights into our model's proficiency in handling extended dialogues across a broad spectrum of subjects.

### 4.2 MoAA Supervised Fine-tuning Results

**MoAA SFT significantly improves model alignment**  As shown in Table 1, applying SFT with our MoA synthetic generated data significantly improves performances on both models. After SFT, Llama-3.1-8B-Instruct's win rate for both AlpavalEval 2 and Arena-Hard roughly doubled against GPT-4 baselines. MT-Bench also achieves significant performance gains (8.01 vs. 8.33, maximum score is 10.0) despite the scores of MT-Bench being more saturated than others. Improvements on Gemma-2-9b-it is still significant albeit to a lesser degree. We posit this to be the Gemma family being heavily distilled already on these benchmarks considering their original high benchmark scores. We observed a 6.36 and 5.6 points increase from the original model for AlpacaEval 2 and Arena-Hard respectively. Note that our two-layer MoA framework *MoA-Data-Synthesis* achieves impressive performance across all benchmarks, contributing to the high SFT results. These consistent and significant improvements highlight the robustness and effectiveness of MoAA.

**Selection of instruction datasets matters**  Table 2 illustrates the influence of instruction tuning set compositions on model performance. We evaluated three configurations: Ultrafeedback (UF), Ultrachat (UC), and a combination of the two (UF + UC). The Ultrafeedback dataset comprises roughly 61,000 training instructions, while from the larger Ultrachat dataset of 200,000 instructions, we subsampled 60,000 to maintain scale parity with Ultrafeedback. The combined set, UF + UC, integrates all Ultrafeedback instructions with an additional 5,000 from Ultrachat.

Our findings reveal that the combined UF + UC dataset generally yields the highest performance across both Llama and Gemma models. It closely matches or marginally trails the Ultrachat set in some benchmarks while outperforming it in others. The Ultrafeedback set, while the least effective overall, demonstrates efficacy in the Arena-Hard benchmark. Notably, the Ultrachat set enhances performance on MT-Bench, likely due to its inclusion of multi-turn conversational data. It's important to note that this analysis does not represent an exhaustive search for the optimal instruction set combination. We posit that a more meticulous selection of datasets, encompassing diverse domains and difficulty levels, could further enhance SFT performance.

Table 2: The influence of instruction tuning set compositions on the model performance. We pick three different sets: Ultrafeedback (UF), Ultrachat (UC), and a mixture of the two (UF + UC). Ultrafeedback has roughly 61,000 data points. Ultrachat we sampled 60,000 data points. And for the mixture, we include all Ultrafeedback data and 5,000 Ultrachat samples.

| Model | MOAA Data | AlpacaEval 2 (LC) | MT-Bench | Arena-Hard |
|---|---|---|---|---|
| | Ultrafeedback | 39.92 | 8.10 | 39.8 |
| Llama-3.1-8B-Instruct | Ultrachat | **43.86** | **8.39** | 39.5 |
| | UF + UC | 43.77 | 8.33 | **40.8** |
| | | | | |
| | Ultrafeedback | 51.56 | 7.88 | 45.4 |
| Gemma-2-9B-it | Ultrachat | 51.43 | **8.67** | 45.1 |
| | UF + UC | **53.79** | 8.65 | **47.6** |

Table 3: Model performances by SFT on the data generated by single models and MoA. All models are tuned on the original Llama-3.1-8B-Instruct. The **Teacher** column indicates the model used to generate the data for SFT. We use UF + UC as the dataset for all experiments.

| Model | Teacher | AlpacaEval 2 (LC) | MT-Bench | Arena-Hard |
|---|---|---|---|---|
| | *N/A (No SFT Reference)* | 22.33 | 8.01 | 19.5 |
| | *N/A (Original Data)* | 14.50 | 7.73 | 11.7 |
| Llama-3.1 | Llama-3.1-70B-Instruct | 14.53 | 7.84 | 10.4 |
| -8B-Instruct | Qwen2-72B-Instruct | 20.50 | 7.88 | 19.5 |
| | Llama-3.1-405B-Instruct | 24.26 | 8.06 | 25.2 |
| | Gemma-2-27B-it | 36.86 | 8.12 | 31.4 |
| | Wizardlm-2-8x22B | 33.26 | 8.44 | 36.5 |
| | GPT-4o-05-13 | 33.95 | **8.55** | 36.9 |
| | MoAA-SFT | **43.77** | 8.33 | **40.8** |

**Superior quality of MoAA synthesized data**   We conducted an ablation study comparing SFT performances using data synthesized by our method against that generated by single models. Table 3 presents the results that models fine-tuned using our MoAA data synthesis approach outperform those trained on data from individual open-source models.

To further underscore the advantages of our method, we extended our comparison to include data generated by `GPT-4o-05-13`, one of the most powerful closed-source models currently available. Notably, models fine-tuned on our synthesized data demonstrate superior performance benchmarks compared to those trained on `GPT-4o-05-13` data, with the exception being MT-Bench. The strength of our approach is particularly noteworthy given that our method exclusively utilizes open-source models. Note that MoA can incorporate closed-source model to further improve performance (Wang et al., 2024c). Further exploration on this can be pursued in future work.

**Effectiveness of MoA Architecture over Naive Model Mixtures**   To validate the efficacy of our Mixture of Agents (MoA) architecture and distinguish it from simple multi-model aggregation, we conducted an ablation study comparing MoA against two naive mixture approaches and one approach that utilizes one state-of-the-art reward model to pick the best response. The first approach, which we term "Combined 5," combined all datasets labeled by the five LLMs used in our MoA setup. Specifically, each LLM will generate responses for the entire dataset and we combine all of them into one big SFT set that is five times the original size. The second approach, term "Random 5," randomly sampled one response for each instruction from these five models and maintained the same data size. Lastly, "Best of 5" uses a strong reward model ArmoRM-Llama3-8B-v0.1 (Wang et al., 2024a) to rank responses from these five models and pick the best one as the training response. For multi-turn data, we average the score of each turn for each conversation.

As illustrated in Figure 1 and detailed in Table 4, both naive mixture methods significantly underperform our MoA approach across all three benchmarks. This substantial performance gap underscores that MoA's success is not merely a result of utilizing multiple models. The "Best of 5" method, while marginally better on MT-Bench, underperforms MoA on AlpacaEval2 and Arena-

Table 4: Model performances by SFT on data generated by baseline multi-model methods and MoA. We finetune on Llama-3.1-8B-Instruct with the same training setups as MoA-SFT including the dataset. Combine 5: including all five responses generated by each individual model. Random 5: random sampling of one response from the five models for each instruction. Best of 5: choosing the best response out of five models for each instruction using ArmoRM-Llama3-8B-v0.1.

| Method | AlpacaEval 2 (LC) | MT-Bench | Arena-Hard |
|---|---|---|---|
| Llama-3.1-8B-Instruct | 22.33 | 8.01 | 19.5 |
| Combined 5 | 27.23 | 8.17 | 26.7 |
| Random 5 | 25.30 | 8.19 | 26.4 |
| Best of 5 | 35.62 | **8.36** | 38.6 |
| Llama-3.1-8B-Instruct-MoAA-SFT | **43.77** | 8.33 | **40.8** |

Table 5: Performance comparison of the MoA reward model and other widely-used reward models on Rewardbench.

| Model Type | Method/Model | Chat | Chat Hard | Safety | Reasoning | Average |
|---|---|---|---|---|---|---|
| Open-Source | Llama-3.1-70B-Instruct | **97.2** | **70.2** | 82.8 | 86.0 | 84.0 |
| | Gemma-2-27B-it | 94.8 | 59.1 | 86.4 | 83.3 | 80.9 |
| | Qwen2-72B-Insutrct | 96.2 | 64.6 | 86.0 | 86.1 | 83.2 |
| | MoA as reward model | 94.7 | 69.4 | **90.6** | **87.7** | **85.6** |
| Fine-Tuned | ArmoRM-Llama3-8B-v0.1 | 96.9 | 76.8 | 90.5 | 97.3 | 90.4 |
| | PairRM | 90.2 | 52.2 | 47.7 | 49.0 | 59.8 |
| Closed-Source | GPT-4o-2024-05-13 | 96.6 | 70.4 | 86.5 | 84.9 | 84.6 |

Hard. Despite ArmoRM-Llama3-8B-v0.1 being a state-of-the-art reward model and top-scoring on the RewardBench, our MoA approach performs better on average. These results demonstrate that our architecture goes beyond simple aggregation, organically combining and refining proposer responses to generate high-quality data.

### 4.3 MoAA Preference Alignment Results

**MoAA DPO improves model alignment further** To further enhance model alignment, we align our SFT models with a widely used preference optimization method called direct preference optimization (DPO). Models tuned by DPO on our MoA preference alignment dataset (termed *MoAA-DPO* at the end) outperforms MoAA-SFT tuning significantly on all three benchmarks, for both Llama and Gemma models, as evidenced in Table 1.

**MoA as a Reward Model: Comparison with State-of-the-Art** To assess the effectiveness of our MoA reward model, we conducted a comparison against state-of-the-art reward models and open-source generative-LLM-based reward model. On RewardBench, our MoA method demonstrates a clear improvement, achieving a 1.6 point increase over the best open-source model incorporated in our MoA setup, as illustrated in Table 5. It is especially effective at the Safety category, scoring 4.2 points higher than the highest open-source model incorporated in MoA. It is noteworthy that this performance gain is achieved without any specific tuning for reward modeling, underscoring the inherent strength of our MoA method.

Surprisingly, despite scoring lower than ArmoRM on RewardBench, the model DPO-tuned on our MoA preference alignment dataset exhibits highly competitive performance shown in Table 6. It outperforms ArmoRM-tuned models on both MT-Bench and Arena-Hard benchmarks, with only a marginal deficit on AlpacaEval2. Furthermore, our method outperforms individual LLMs used as components within the MoA structure when these are employed as standalone reward models. This observation reinforces the synergistic benefit of our MoA architecture, demonstrating its ability to leverage the collective strengths of multiple models effectively.

**Ablation Study on MoA Alignment Paradigms** We conducted an extensive exploration of alternative approaches to utilize the MoA framework during Stage 2 of our alignment process. Two

Table 6: Performance comparison of models using different reward models. All settings generate five candidate responses with a temperature of 0.8 and use the reward model to pick chosen and rejected responses as the preference pair. We use the same *Llama-3.1-8B-Instruct-SFT* as the base model for DPO across all setups.

| Reward Model | AlpacaEval 2 (LC) | MT-Bench | Arena-Hard |
|---|---|---|---|
| Llama-3.1-70B-Instruct | 55.35 | 8.36 | 45.1 |
| Qwen2-72B-Instruct | 55.80 | 8.31 | 43.5 |
| Gemma-2-27B-it | 56.81 | 8.31 | **48.8** |
| GPT-4o-2024-0806 | 55.05 | **8.76** | 44.1 |
| MoA as reward model | **57.23** | 8.58 | 48.3 |
| | | | |
| ArmoRMLlama3-8B-v0.1 | **57.79** | **8.56** | **42.3** |
| PairRM (Jiang et al., 2023b) | 50.17 | 8.33 | 42.2 |
| | | | |
| *N/A (SFT Reference)* | *43.77* | *8.33* | *40.8* |

additional primary variants were investigated: *MoA-OnPolicy* and *MoA-OffPolicy*. In the *MoA-OnPolicy* approach, we incorporated the MoAA-SFT model from Stage 1 as the aggregator in an MoA setup. We use the same proposers as in Stage 1 and the MoAA-SFT model as the aggregator to generate candidate responses. Conversely, the *MoA-OffPolicy* method utilized the identical MoA architecture (including the aggregator) from Stage 1 to generate candidate responses, with the same reward model selecting preference pairs. Both settings generate five candidate responses with a temperature of 0.8 and use ArmoRM-Llama3-8B-v0.1 as the reward model. The preference pairs were then selected using the ArmoRM-Llama3-8B-v0.1 reward model.

The results of this ablation study, as presented in Figrue 4, reveal insights into the efficacy of these approaches. The *MoA-OffPolicy* method demonstrated lower performance scores, which can be attributed to a potential distribution mismatch between the generated data and the model, as the responses were not directly generated by the SFT model. While *MoA-OnPolicy* leveraged the SFT model as an aggregator to generate "on-policy" data, it failed to exhibit the anticipated benefits of the MoA structure in this context. We hypothesize that this limitation stems from the SFT model's training as a response generator rather than an aggregator designed to combine and refine responses. Collectively, these findings provide evidence that the MoA framework is more effectively employed as a reward model during the DPO stage.

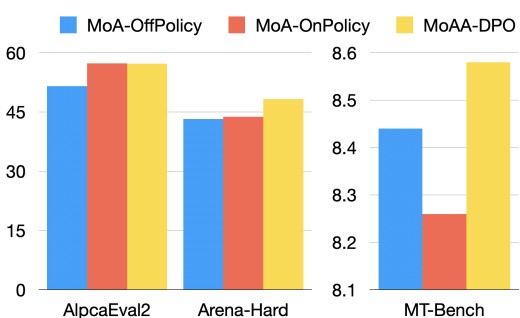

Figure 4: Performance comparison of models using different DPO settings. *MoA-OnPolicy* uses the SFT model to generate on-policy responses in a MoA style, with the SFT model as the aggregator and unchanged proposers. *MoA-OffPolicy* uses the MoA architecture in stage 1 to generate responses.

## 5 CONCLUSION

This paper presents Mixture of Agents Alignment, a model alignment recipe that leverages multiple LLMs' expertise at the two stages of the alignment process. By harnessing the collective intelligence of open-sourced LLMs, MoA is proven to be a powerful synthetic data generator during the SFT stage, and a competitive reward model during DPO. Models fine-tuned on our MoA generated synthetic data achieves significant improvement on evaluation benchmarks such as AlpacaEval 2, MT-Bench, and Arena-Hard. Utilizing our MoA as a reward model with criteria filtering also proves to be able to produce competitive models compared to DPO models using state-of-the-art reward models. Extensive ablation studies demonstrate the efficacy and careful design of our MoAA strategy.

## 6 REPRODUCIBILITY STATEMENT

We put considerable effort into ensuring our results, models, and datasets are reproducible. We included code and data in our supplementary material. Additionally, Section 4.1 as well as Appendix A details the models, datasets, and hyperparameters we used to arrive at the results shown in the paper.Section 3 and Section 4.1 details the methodology and specific design choices of our MoA approach. Tables 18 to 21 list the exact prompts used by our method.

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

## A Hyperparameters

**SFT hyperparameter settings**   For both the Llama model and Gemma model, we use learning rate of $8.0e$-7 and gradient accumulation of 128. For the Llama model we train for 6 epochs whereas for Gemma we train for 5 epochs. All experiments are done on one node of 8xA100.

**DPO hyperparameter settings**   Hyperparameters are crucial for preference optimization methods. For Llama model, we use learning rate of $8.0e$-7. For Gemma model, we use a learning rate of $3.0e$-7. For both setups, we train for 5 epochs with a beta of 0.01 and gradient accumulation of 128. All experiments are done on one node of 8xA100.

## B MoA architecture selection

**MoA architecture for Stage 1 data synthesis**   We use a two-layer MoA framework with WizardLM-8x22B, Qwen2-72B-Instruct, Gemma-2-27B-it, LLaMA-3.1-70B-Instruct as proposers and Qwen1.5-110B-Chat as the aggregator. This specific choice is based on insights from previous work (Wang et al., 2024c) and some empirical search. Specifically, previous work has shown that WizardLM-8x22B is a great proposer whereas Qwen1.5-110B-Chat is a great aggregator. Then we just add strong open-source models that have decent performances such as Qwen2-72B-Instruct, Gemma-2-27B-it, and LLaMA-3.1-70B-Instruct as proposers to get our final architecture. We have tried a bunch of other setups, e.g., using only three proposers, or using Qwen2-72B-Instruct, Gemma-2-27B-it, or LLaMA-3.1-70B-Instruct as the aggregator. Even though the current setup as shown in Table 7 doesn't yield the highest performance out of other setups, it is the most balanced across three benchmarks. Note that a more explicit and intelligent search method can be used to find potentially better architecture. We leave this interesting exploration to future work. To balance efficiency and performance, we set the number of layers to two. Our model pool is limited to the most capable general-purpose models available at the time, ensuring broad generalization, while domain-specific fine-tuned models (e.g., for code) were not included. Regarding the robustness of ensemble composition, an early observation was that the order of proposers has minimal impact, so we generally arrange them from strongest to weakest.

**MoA architecture for Stage 2 preference ranking**   We select our architecture in a similar manner during this stage. Notably, Qwen2-72B-Instruct appears to be a better aggregator at evaluating model responses than others. Hence after some empirical search, the MoA architecture has proposers including Gemma-2-27B-it, LLaMA-3.1-70B-Instruct, Qwen2-72B-Instruct, and Qwen2-72B-Instruct as the aggregator.

Table 7: Performance of different MoA architecture. WGQL stands for those four models: WizardLM-8x22B, Qwen2-72B-Instruct, Gemma-2-27B-it, LLaMA-3.1-70B-Instruct. WGQ stands for the first three models shown before.

| Aggregator | Proposers | AlpacaEval 2 (LC) | MT-Bench | Arena-Hard |
|---|---|---|---|---|
| Qwen2-72B-Instruct | WGQL | 59.81 | 9.19 | 79.3 |
| Gemma-2-27B-it | WGQL | 63.47 | 9.19 | 70.8 |
| LLaMA-3.1-70B-Instruct | WGQL | 45.30 | 9.29 | 70.8 |
| Qwen1.5-110B-Chat | WGQ | 61.80 | 8.93 | 76.4 |
| Qwen1.5-110B-Chat (chosen) | WGQL | 62.50 | 9.17 | 75.9 |

**Can we automatically search for an architecture?**   To be more efficient than conducting a manual sweep, we did an early investigation on whether we can use an automatic optimization pipeline to find a good LLM mixture. We will include some details on how we do that here.

Setup: Specifically, we fix the number of layers to be two and the aggregator to be Qwen-1.5-110b-Chat, and set the number of models and which model in proposers to be variables for optimization. We utilized Broyden–Fletcher–Goldfarb–Shanno algorithm (BFGS) for this unconstrained optimization problem. We use the LLMs used in the original MoA-Lite from Wang et al. (2024b) as

Table 8: Performance comparison of MoA-Lite and MoA searched using our proposed optimization method. Note that this MoA-Lite mixture is taken from the original MoA paper and has lower performances than our mixture.

| Model | Aggregate | AlpacaEval (LC) | Arena-Hard | MT-Bench |
|---|---|---|---|---|
| MoA-Lite | 74.1 | 59.3 | 71.3 | **9.18** |
| MoA-Lite–searched | **75.0** | **62.0** | **71.8** | 9.11 |

a starting point. This means the MoA has Qwen-1.5-110b-Chat as aggregator and Qwen1.5-110B-Chat (Bai et al., 2023b), Qwen1.5-72B-Chat, WizardLM-8x22B (Xu et al., 2023a), LLaMA-3-70B-Instruct (Touvron et al., 2023b), Mixtral-8x22B-v0.1 (Jiang et al., 2024), dbrx-instruct (The Mosaic Research Team, 2024) as proposers. Note this mixture has a lower score than the mixture we used in this paper.

Validation Data: It is important to have a good set of validation data. We randomly sampled 50 problems from AlpacaEval and 50 from Arena-Hard. The combined size of 100 enables us to verify architecture performances quickly. We averaged the scores of AlpacaEval and ArenaHard to be our final metric.

We ran the optimization and found the best mixture to be WizardLM-2-8x22b, Qwen-1.5-110b-Chat, Qwen-1.5-72b-Chat, and three Llama-3-70b-Instruct as proposers and Qwen-1.5-110b-Chat as aggregator. The resulting mixture outperforms our MoA-Lite on two out of the three benchmarks as shown in Table 8.

## C  COST EFFICACY OF MoA

**Data generation cost**  In this section, we compare the cost efficacy of our MoA data generation process vs using a strong closed-source model such as GPT-4o-05-13. To make this a fair comparison, we measure the cost of generating synthetic data using Ultrafeedback for both MoA and GPT-4o-05-13. MoA requires around $365.9 whereas GPT-4o-05-13 requires $429.4 as demonstrated in Table 9. MoA saves about 23% and achieves much higher performance. The MoA cost is computed using the cost detailed on Together Endpoint and the GPT-4o-05-13 cost is taken from their website.

Table 9: Cost comparison across models for generating instruction tuning dataset. MoA saves 23% of the cost compared to GPT-4o-05-13 while achieves higher performance shown in Table 6

| Model | $ per Million Tokens | Cost to Generate Dataset |
|---|---|---|
| Qwen1.5-110B-Chat | 1.8 | - |
| WizardLM-2-8x22B | 1.2 | 55.53 |
| Llama-3-70b-Instruct | 0.9 | 30.07 |
| Qwen2-72B-Instruct | 0.9 | 25.12 |
| gemma-2-27b-it | 0.8 | 23.85 |
| Gemma-2-9B-it-MoAA-DPO | 0.3 | - |
| MoA | 5.6 | **365.95** |
| gpt-4o-2024-05-13 | 7.5 | 429.45 |

**Inference efficiency of MoAA**  One of the key motivations for developing MoAA is to address the practical limitations of using MoA for cost/latency-sensitive scenarios. Compared to standalone LLMs, deploying MoA at inference time is computationally expensive and incurs high latency due to the need to generate and aggregate responses from multiple large models. This motivates us to align its knowledge to a smaller standalone model, while ensuring that the MoAA-trained model retains response quality comparable to the aggregated outputs of MoA. In our inference efficiency analysis in Table 10, Gemma-2-9B-it-MoAA-DPO achieves 90.6% of the MoA performance with only 5.4% of the cost of MoA.

Table 10: Inference efficiency analysis comparison of our methods and MoA. We show that with only 5.4% of the cost of MoA, our method can achieve 90.6% of the MoA performance.

| | AE (LC) | AH | MT-Bench | Avg. | % of MoA | $/M tokens |
|---|---|---|---|---|---|---|
| Gemma-2-27b-it | 52.3 | 52.3 | 8.86 | 64.4 | 83.9% | 0.8 |
| Llama-3-70b-Instruct | 37.3 | 55.2 | 8.99 | 60.8 | 79.3% | 0.9 |
| Qwen2-72B-Instruct | 38.1 | 45.0 | 8.88 | 57.3 | 74.7% | 0.9 |
| WizardLM-2-8x22B | 51.3 | 71.3 | 8.78 | 70.1 | 91.4% | 1.2 |
| Qwen1.5-110B-Chat | 43.9 | 56.4 | 8.96 | 63.3 | 82.5% | 1.8 |
| | | | | | | |
| Llama-3.1...-MoAA-DPO | 57.2 | 48.3 | 8.58 | 63.8 | 83.2% | 0.2 |
| Gemma-2...MoAA-DPO | 63.9 | 55.6 | 8.91 | 69.5 | 90.6% | 0.3 |
| MoA | 62.5 | 75.9 | 9.17 | 76.7 | 100% | 5.6 |

# D    REASONING EVALUATIONS

We conducted extensive testing on math, coding, knowledge, and complete reasoning benchmarks. The datasets evaluated include MMLU (Hendrycks et al., 2020), HumanEval (Chen et al., 2021) and GPTQA (Rein et al., 2023) and MATH (Hendrycks et al., 2021).Even though we did not explicitly add any of those data in our instruction dataset or preference alignment dataset, we want to verify if the model tuned can generalize to other domains and not just overfit to the tuning set. In Table 11, we observed a slight decrease in math, reasoning, and coding ability during SFT with MoAA, followed by recovery during the DPO stage. Notably, for Gemma, the model fine-tuned with MoAA outperforms the original model in overall performance. This means our tuned model remain fairly robust and generalize to challenging reasoning tasks despite not having any explicit reasoning data added. Composing a more balanced dataset mixture with reasoning data is a nice direction of future work.

Table 11: Reasoning evaluations of different models across MMLU, HumanEval , GPQA, MATH.

| Model | MMLU | HumanEval (pass@1) | GPQA | MATH | Average |
|---|---|---|---|---|---|
| Llama-3.1-8B-Instruct | **0.7089** | **0.6671** | 0.2273 | **0.51** | **0.527** |
| Llama-3.1-8B-Instruct-MoAA-SFT | 0.6854 | 0.5793 | 0.2626 | 0.48 | 0.502 |
| Llama-3.1-8B-Instruct-MoAA-DPO | 0.6864 | 0.5354 | **0.3434** | 0.49 | 0.514 |
| | | | | | |
| Gemma-2-9B-it | **0.7382** | **0.6341** | 0.2929 | 0.50 | 0.541 |
| Gemma-2-9B-it-MoAA-SFT | 0.7356 | 0.6085 | 0.2828 | **0.52** | 0.537 |
| Gemma-2-9B-it-MoAA-DPO | **0.7382** | 0.6329 | **0.3081** | **0.52** | **0.549** |

# E    ADDITIONAL BASELINES

In this section, we present a comparison with several additional baselines to strengthen the effectiveness of our method. Specifically, we compare with

- MagPie (Xu et al., 2024), a contemporary method that follows a similar SFT and DPO process with its generated data.

- Meta-Rewarding LLM (Wu et al., 2024), an iterative alignment method that utilizes self-judgment to self-improve.

- Original Ultrafeedback (contains 61135 data points) + same 5000 data subsampled from Ultrachat

- MOAA-SFT Ultrafeedback samples (contains 60000 data points) and MoAA-DPO on same 6000 Ultrafeedback data.

As shown in Table 12 and Table 13, our Llama-3.1-8B-Instruct-MoAA-DPO achieves competitive performance compared to all the baselines above, demonstrating the effectiveness of our approach.

Because both MagPie and Meta-Rewarding LLMs are built based on Llama-3, we tuned a Llama-3-8B-Insutrct with MoAA-SFT to compare. Our approach still show stronger performances.

Table 12: Comparison of our method and MagPie and Meta-Rewarding LLM on AlpacaEval and Arena-Hard. MagPie's result was taken directly from the paper. Our method achieves superior performance on both benchmarks. For Meta-Rewarding LLM, we selected the scores from the last iteration (iteration 4) which is the highest in the paper.

| Model | Base Model | Data Size | AlpacaEval (LC) | Arena-Hard |
|---|---|---|---|---|
| Llama-3-8B-Instruct | - | - | 24.01 | 20.6 |
| Llama-3.1-8B-Instruct | - | - | 26.06 | 28.0 |
| MAGPIE-Pro-SFT | Llama-3-8B-Base | 300k | 25.08 | 18.9 |
| MAGPIE-Pro-DPO | MAGPIE-Pro-SFT | 100k | 50.10 | 25.7 |
| Meta-RewardingLM Iter4 | - | - | 39.44 | 29.1 |
| Llama-3...MoAA-SFT | Llama-3-8B-Instruct | 61k+5k | 42.61 | 31.9 |
| Llama-3.1...MoAA-SFT | Llama-3.1-8B-Instruct | 61k+5k | 43.77 | 40.8 |
| Llama-3.1...MoAA-DPO | Llama-3.1...MoAA-SFT | 6k | 57.23 | 48.3 |
| Gemma-2...MoAA-DPO | Gemma-2...MoAA-SFT | 6k | **63.75** | **55.6** |

Table 13: Performance metrics of two other baseliens. 1) Llama-3.1-8B-Instruct tuned on the original responses from Ultrafeedback and Ultrachat. 2) MoAA-SFT on a 60,000 subsample of Ultrachat. Here we chose sample size to be 60,000 because we want to maintain a similar data scale to our original MoAA-SFT setup. Then we perform MoAA-DPO with the same setup as the original MoAA-DPO in the paper, using the same 6,000 Ultrafeedback data, but generated on policy with the Ultrachat SFT model.

| Model | AlpacaEval2 (LC) | Arena Hard | MT-Bench |
|---|---|---|---|
| SFT on Ultrachat and Ultrafeedback | 14.50 | 11.7 | 7.73 |
| Llama-3.1-8B-Instruct-MoAA-SFT | **43.77** | **40.8** | **8.33** |
| Llama-3.1-8B-Instruct-MoAA-SFT (UC) | 43.86 | 39.5 | 8.39 |
| Llama-3.1-8B-Instruct-MoAA-DPO (UC) | **58.15** | **42.6** | **8.64** |

## F STRENGTHENING THE STRONGEST MODEL IN MOA

In this section, we tried to answer the question of whether our method can scale when the strongest model in the mix was trained rather than a much weaker model. It turned out we still observed a clear performance boost with MoA alignment. We think this is a non-trivial finding because improving the strongest model in the mix provides evidence that our method can potentially push the frontier open-source models further without the supervision of stronger LLMs. Specifically, we evaluated a small-scale MoA setup with Gemma-2-9B-it, Llama-3.1-8B-Instruct, and Mistral-7B-Instruct-v0.3 (Jiang et al., 2023a) as proposers, and used a two-layer MoA with Gemma-2-9B-it as the aggregator to generate the data mix.

In Table 14, the fine-tuned Gemma model shows better performance than the strongest individual model (itself) in the mix by a large margin. This is s very promising result since we are improving LLMs to be better than the teachers.

We also provide a study on the performances of this MoA architecture in Table 15. We see that performances in general increase with the increase of layers, although the plateau is starting to occur.

Table 14: Performance of Gemma-2-9b-it model fine-tuned by small-scale MoA setup. We can see that it actually outperforms the best individual model that comprised the MoA.

| Model | AlpacaEval (LC) | AlpacaEval | Arena-Hard | MT-Bench |
|---|---|---|---|---|
| Mistral-7B-Instruct-v0.3 | 19.88 | 15.67 | 16.3 | 7.59 |
| Llama-3.1-8B-Instruct | 26.06 | 27.48 | 28 | 8.34 |
| Gemma-2-9b-it | 48.54 | 36.26 | 40.6 | 8.49 |
| **SFT on Gemma-2-9b-it** | **54.19** | **44.99** | **44** | **8.78** |

Table 15: Model performances of small-scale MoA across different models as final aggregator.

| Aggregator | Layer | AlpacaEval2 (LC) | AlpacaEval2 | Arena-Hard | MT-Bench |
|---|---|---|---|---|---|
| Gemma-2-9b-it | 2 | **56.62** | 47.91 | 48.1 | 8.63 |
| | 3 | 55.75 | **48.72** | **51.0** | **8.65** |
| Llama-3.1-8b-Instruct | 2 | 30.73 | 39.47 | 36.4 | 8.16 |
| | 3 | 30.06 | 39.55 | 38.3 | 8.33 |
| Mistral-7b-instruct-v0.3 | 2 | 26.75 | 24.55 | 25.4 | 8.01 |
| | 3 | 29.97 | 29.55 | 29.4 | 8.38 |

# G    MORE MoA AS A REWARD MODEL EVALUATION

In this section, we provide additional benchmarking on MoA as a reward model on the PPE benchmark (Frick et al., 2024). PPE consists of 18k diverse data points spanning human preference and reasoning tasks. Table 16 show that MoA as a reward model outperforms the best individual model in its mix by a significant margin and also exceeds GPT-4o-mini in overall performance. Compared to Skywork-Reward-Gemma-2-27b, which scores 9 points higher on the Reward Bench, MoA achieves 9.5 points higher on the PPE benchmark. We believe this performance difference highlights an issue with the Reward Bench: it has become overspecialized due to fine-tuning efforts since its launch, making fine-tuned models appear more capable than they actually are. PPE, as a newer and more diverse benchmark, provides a clearer evaluation of model capabilities and further demonstrates the effectiveness of MoA as a robust reward model.

Table 16: Our MoA as reward model's performance on PPE, compared with other LLM as a judge and reward model.

| Model | MMLU Pro | MATH | GPQA | MBPP Plus | IFEVAL | Human Pref. | AVG |
|---|---|---|---|---|---|---|---|
| MoA as reward model | 0.76 | 0.79 | 0.58 | 0.62 | 0.57 | 0.6465 | 0.661 |
| Qwen-2-72b-Instruct | 0.72 | 0.73 | 0.56 | 0.58 | 0.54 | 0.6135 | 0.624 |
| Llama-3.1-70b-Instruct | 0.73 | 0.73 | 0.56 | 0.58 | 0.56 | 0.6429 | 0.634 |
| Gemma-2-27b-it | 0.68 | 0.73 | 0.54 | 0.58 | 0.52 | 0.6169 | 0.611 |
| GPT-4o-mini-2024-07-18 | 0.71 | 0.81 | 0.57 | 0.54 | 0.56 | 0.6646 | 0.642 |
| Claude-3.5-Sonnet-20240620 | 0.81 | 0.86 | 0.63 | 0.54 | 0.58 | 0.6733 | 0.682 |
| Skywork-Reward-Gemma-2-27b | 0.54 | 0.63 | 0.53 | 0.59 | 0.54 | 0.5662 | 0.566 |
| ArmoRM-Llama3-8B-v0.1 | 0.66 | 0.71 | 0.57 | 0.54 | 0.58 | 0.6057 | 0.610 |

# H GENERALIZATION TO OTHER ARCHITECTURE AND MODEL SIZE

To verify if our method can generalize to other architecture or model sizes, we fine-tuned a Llama-3.2-3b-Instruct using our MoAA-SFT pipeline. Llama-3.2 is the newest model in the Llama family at the point of writing. In addition, we picked the size to be 3B to verify if it would work on smaller LLMs. Table 17 shows the result of our MoAA-SFT. We found convincing improvements on all three benchmarks. Possibly due to model size, the improvements are not as big as what we saw in 8b/9b models. Nonetheless, our method is able to train a very competitive 3B LLM.

Table 17: Performance Comparison of Llama-3.2-3b Model fine-tuned on MoAA-SFT.

| Model | AlpacaEval (LC) | Arena-Hard | MT-Bench |
|---|---|---|---|
| Llama-3.2-3b-Instruct | 19.9 | 14.2 | 7.64 |
| Llama-3.2-3b-Instruct-MoAA-SFT | **35.4** | **21.9** | **8.11** |

# I PROMPT TEMPLATES

Table 18: Aggregate-and-Synthesize Prompt to integrate responses from other models.

You have been provided with a set of responses from various open-source models to the latest user query. Your task is to synthesize these responses into a single, high-quality response. It is crucial to critically evaluate the information provided in these responses, recognizing that some of it may be biased or incorrect. Your response should not simply replicate the given answers but should offer a refined, accurate, and comprehensive reply to the instruction. Ensure your response is well-structured, coherent, and adheres to the highest standards of accuracy and reliability.

Responses from models:
1. [Model Response from $A_{i,1}$]
2. [Model Response from $A_{i,2}$]
...
$n$. [Model Response from $A_{i,n}$]

Table 19: Prompt to select evaluation criteria for responses from reward modeling.

Analyze the following user query and two AI assistant responses. Your task is to determine the three most relevant evaluation criteria for assessing these responses. Choose exactly 3 criteria from the list below that are most applicable to this specific query and responses:

1. Instruction adherence: How well the response follows the user's instructions.
2. Relevance: How directly the response addresses the user's query.
3. Accuracy: The correctness and up-to-date nature of the information provided.
4. Depth: The comprehensiveness and level of detail in the answer.
5. Clarity: How well-structured and easy to understand the response is.
6. Helpfulness: How useful the response is in solving the user's problem or answering their question.
7. Safety: How well the response handles potentially sensitive or dangerous requests.
8. Robustness: How well the response handles nuanced or ambiguous aspects of the query.

Here's an example to guide your selection and output formatting:
Example User Query: "What are the health benefits of drinking green tea?"
Example Assistant A Response: "Green tea has many health benefits. It contains antioxidants that can improve brain function and fat loss. It may also lower the risk of certain cancers and cardiovascular diseases."
Example Assistant B Response: "Green tea is good for you. It has stuff that helps your brain and makes you lose weight. It might also stop you from getting sick."

Example Output:
Selected Criteria:
1. Accuracy
2. Depth
3. Clarity

Explanation: For this query about health benefits of green tea, accuracy is crucial to ensure the information provided is correct. Depth is important to cover the range of potential benefits comprehensively. Clarity is necessary to ensure the information is presented in an understandable manner, especially when dealing with scientific health information.

Now, please analyze the following actual query and responses:
User query: {question}
Assistant A response: {answer_a}
Assistant B response: {answer_b}

Output your selected criteria strictly using the following format:
Selected Criteria:
1. [Criterion 1]
2. [Criterion 2]
3. [Criterion 3]

Explanation: [Briefly explain why you chose these three criteria]

Table 20: Proposer prompt for reward modeling.

As an impartial expert evaluator, your task is to critically assess the responses provided by two AI assistants (A and B) to a user query. Follow these steps:

1. Understand the Query: Carefully analyze the user's question or request to grasp its specific nature and requirements.

2. Criteria: Focus your evaluation on these three criteria. For each criterion, provide a brief assessment of how well each assistant performed, and then compare them directly.
{criteria}

3. Evaluation: For each selected criterion, provide a qualitative assessment using natural language. Consider using the following phrases:
    - Exceptional
    - Strong
    - Satisfactory
    - Needs improvement
    - Inadequate

4. Evaluation Process:
- Provide assessment and brief explanation for each criterion
- Summarize key strengths and weaknesses of each response
- Comparative Analysis:
    - Compare the overall performance of both responses
    - Explain your reasoning process, referring to specific aspects of each response
    - Do not let factors such as response length, assistant names, or the order of presentation influence your decision

Table 21: Aggregator prompt for reward modeling.

As an expert meta-evaluator, your task is to analyze and synthesize multiple evaluations comparing two AI assistants' responses (A or B) to a user query. Your role is crucial in determining the final assessment. Please consider the following:

1. Assess the consistency and validity of arguments across all evaluations.
2. Identify any potential biases, errors, or oversights that may have influenced individual evaluations.
3. Consider the strengths and weaknesses of each AI response as highlighted across all evaluations.
4. Synthesize a final, comprehensive evaluation that:
    a) Provides a clear comparison of the two AI responses.
    b) Addresses any conflicting opinions among the evaluations.
    c) Offers a well-reasoned, definitive judgment on which response better addresses the user query.
    d) Strictly using "[[A]]" if assistant A is better, or "[[B]]" if assistant B is better to indicate your preferred response.

Do not let factors such as response length, assistant names, or the order of presentation influence your decision.

The evaluation should be based on the following criteria:
{criteria}

User query: {question}
Assistant A response: {answer_a}
Assistant B response: {answer_b}

Individual evaluations:
{proposer_evaluations}

Final Meta-Evaluation:

Table 22: Performance comparison of MoA with and without criteria filtering on Rewardbench.

| Method | Chat | Chat Hard | Safety | Reasoning | Average |
|---|---|---|---|---|---|
| MoA without Filtering | **95.5** | 68.8 | 88.1 | 85.6 | 84.5 |
| MoA with Filtering | 94.7 | **69.4** | **90.6** | **87.7** | **85.6** |

