# OpenReview forum: "Improving Model Alignment Through Collective Intelligence of Open-Source Models"
_ICLR.cc/2025/Conference — Submitted to ICLR 2025_

### Official Review · Reviewer_UnSv · 2024-10-21

**Soundness:** 3
**Presentation:** 2
**Contribution:** 2
**Rating:** 6
**Confidence:** 4

**Summary:**

This paper focuses on creating synthetic data to align LLMs. The authors propose Mixture of Agents Alignment (MoAA), which gathers data using Mixture of Agents (from Wang et al, 2024, which prompts multiple models to propose answers and then aggregate them, showing significant improvement even if the proposed answers are weak/of poor quality on its own), and then finetunes language models such as LLaMA-3.1-8b-Instruct and Gemma-2-8B-it on these data using SFT and DPO.

Specifically, MoAA data collection consists of two stages - SFT data creation and DPO data creation. First, MoAA uses MoA to generate responses from a set of prompts sampled from an instruction-following dataset, and then performs SFT on the generated responses. Next, MoAA uses MoA to judge responses to rate the responses generated by the SFT-tuned models, and then performs DPO on the (highest scoring, lowest scoring) responses. Finally, the authors evaluated this approach in a series of chat benchmarks including Arena-Hard, Alpaca-Eval 2, and MT-Bench, and found sizable improvements compared to other methods such as only using a single model to generate/judge responses.

**Strengths:**

- MoAA presents a competitive method to create high-quality instruction-following data only using open-source models.
- The authors conducted many evaluations and ablation studies about MoAA. Empirical results show that training LMs with MoAA-curated data surpasses other methods such as directly using a single model (e.g., GPT-4o) to generate/judge responses.

**Weaknesses:**

1. This work can be summarized as directly applying MoA to create data, followed by standard SFT and DPO training. This limits its novelty and seems an incremental work based on MoA.

2. Similar to MoA, there are many empirical choices involved in MoAA that could affect its performance and its generalizability. For example, the ensemble of models used for MoAA shows great variance in the end performance (as also discussed in Appendix B). The authors in this work conducted an empirical sweep to determine the models used, but this makes it highly impractical if one wants to extend MoAA in other domains such as math, coding, or agentic tasks.

3. Main results in Table 1 (and similarly in Table 3 and 6) shows significant improvement from MoAA *when comparing* training with MoAA-generated data with prompts from UltraFeedback + UltraChat *against no-training/training with single agent generated data*. However, I believe the presentation (e.g., in Table 1) can be highly misleading, as one natural baseline should be comparing against *directly training on the existing UltraChat and UltraFeedback with SFT/DPO* - a simple setting that has been used in many work and requires no additional data curation. This comparison is necessary to help readers put into perspective the benefit introduced by MoAA, but is right now missing.

4. Since MoAA relies on using multiple large language models (e.g., three 70B+ models for SFT data and also three 70B+ models for DPO data), it becomes necessary to justify this compute-cost-performance trade-off against simply querying proprietary models such as GPT-4o (i.e., in Table 3 and is missing in Table 6). Specifically, in addition to performance, the authors should also compare the inference-cost of using MoAA v.s. GPT-4o to better put this trade-off into perspective.

**Questions:**

The following aspects were unclear in writing:

- Is the subscript in Equation (3) over-loaded? The same subscript notation is used in Equation (1), but from the descriptions, subscripts in Equation (3) is conveying an entirely different meaning than subscripts in Equation (1)?

- Font bolding is used in-consistently in Table 2,4,5. For example, Table 2 showed 8.67 and 8.65 both in bold; Table 4 has 8.36 and 8.33 both in bold; and Table 5 the best performance from last two rows were not in bold (unlike the other tables).


Questions:
- how scalable is the MoAA method? How does performance change if you use more than 2 layers? At which point would one encounter a bottleneck?
- It is unclear whether the method mentioned in Section 3.2.3 is used in the evaluations in Section 4. L256-259 specifically discusses the results of Section 3.2.3, but it was never mentioned in Section 4 again.

---

> ### Comment · Reviewer_UnSv · 2024-11-24
>
> Thank you for the thorough responses and additional empirical results. I believe they have addressed most of my concerns including comparing against simpler methods and the additional compute usage/cost comparison, with a few exceptions (see below). However, I keep my assessment on this work's contribution, since MoAA is mainly a direct application of MoA from prior work.
>
> I have raised my soundness score and overall score, but kept my score on contribution. I detail my follow up questions/concerns below.
>
> ---
>
> > This work can be summarized as directly applying MoA to create data, followed by standard SFT and DPO training. This limits its novelty and seems an incremental work based on MoA.
>
> Thank you for the additional comparison against recent baselines. I agree with these positive results, yet I believe it does not contradict my claim on the contribution: MoAA seems to be a direct application of MoA to create data followed by standard tuning methods. I will increase the soundness score, but I will keep my judgment on contribution.
>
> > Similar to MoA, there are many empirical choices involved in MoAA that could affect its performance and its generalizability. For example, the ensemble of models used for MoAA shows great variance in the end performance (as also discussed in Appendix B). The authors in this work conducted an empirical sweep to determine the models used, but this makes it highly impractical if one wants to extend MoAA in other domains such as math, coding, or agentic tasks.
>
> Thank you for the additional results. Sorry if I misunderstood this, but it seems the "Llama-3.1-8B-Instruct-MoAA-*" results underperforms the original "Llama-3.1-8B-Instruct" for all benchmarks besides gpqa?
>
> > Main results in Table 1 (and similarly in Table 3 and 6) show significant improvement from MoAA when comparing training with MoAA-generated data with prompts from UltraFeedback + UltraChat against no-training/training with single agent generated data. However, I believe the presentation (e.g., in Table 1) can be highly misleading, as one natural baseline should […]
>
> Sorry if my original question was not clear. I intended the authors to compare against performing SFT on UltraChat and DPO on UltraFeedback (instead of SFT on both), which is a common usage of these datasets. Plus, since UltraFeedback is a comparison based dataset where each model response receives a score (and hence many work uses DPO on this), could you clarify how did you perform SFT on it?

---

### Official Review · Reviewer_bGhr · 2024-11-01

**Soundness:** 3
**Presentation:** 3
**Contribution:** 2
**Rating:** 5
**Confidence:** 4

**Summary:**

This paper focuses on improving model alignment through collective intelligence(i.e., Mix-of-Agent) and proposes an alignment method called MoAA. The MoAA consists of two-stage:
(1) the first stage uses MoA as a synthetic data generator to synthesize data for SFT;
(2) the second stage uses MoA as a reward model to align the SFT model with targeted preferences.

By leveraging MoA for data synthesis, this paper does not require costly human annotation and alleviates the potential biases from a single strong model.

Experimental results show that MoAA can improve the win rate of LLaMA-3.1-8B-Instruct from 19.5 to 48.3 on Arena-Hard and from 22.33 to 57.23 on AlpacaEval2, highlighting a promising direction for model alignment through this new scalable and diverse synthetic data recipe.

**Strengths:**

(1) This paper is well-written and has easily understood figures.

(2) The experimental improvement is quite significant.

(3) The idea of using a mix of agents is intuitive and makes sense, as it is expected to improve data quality and reduce biases from a single model.

**Weaknesses:**

1. The technical contributions in this work are relatively limited.  The pipeline in this paper largely relies on existing work (i.e., MoA), merely using its results to construct the preference data for SFT and DPO.

2. Specifically, one-third of the article's methodology section is dedicated to introducing foundational concepts such as DPO and MoA from existing work. Apart from these existing works, the technical differences presented in this paper seem limited to:
  (1) Adjusting more suitable prompts for specific tasks.
  (2) Making some details compatible with multi-turn data.
  (3) Adding a dynamic criteria filtering component in section S3.2. Nevertheless, the ablation results in Appendix Table 12 indicate that the enhancement from criteria filtering is relatively minimal, only around one point. Therefore, the main improvement of this paper still primarily stems from the existing work MoA.

3. I also have some concerns about the comparison of this method with related works.  Although MoAA improves the base model's capabilities, and the authors conducted extensive experiments to test the effectiveness of different teacher models, the comparisons are mainly between MoAA and the base model or other settings of MoAA. There are only Table 5 compares with two other reward models (i.e., ArmRM and PairRM), and MoAA seems to have failed to make a significant difference in the other model. It would be more comprehensive to include comparisons with other similar methods, rather than just analyzing the performance of MoAA under different settings.  At least, compared MoAA to MoA to illustrate the improvement solely brought by MoAA. However, MoA achieve 65.1 and 9.25 in AlpacaEval2(LC) and MT-Bench, which have outperform the performance of MoAA.

4. The experiments in this paper are all based on the small models of 8/9B.  Whether this method can scale to larger models and whether it can lead to significant improvements in larger models is questionable, which may limit the broader impact of this work.

5. Although MoA can enhance performance and reduce bias compared to a single model, it inevitably introduces higher costs for dataset construction.  It is clearly unreasonable to demand that the costs of MoAA be lower than previous methods without using MoA.  However, providing some comparisons in terms of dataset construction efficiency, such as the number of times LLMs are called or consumed tokens, would offer a more comprehensive view. Besides, the related work MoA has given a "Budget and Token Analysis" in Section 3.4.

6. The MoA part uses WizardLM-8x22B, Qwen2-72B-Instruct, Gemma-2-27B-it, and LLaMA-3.1-70B-Instruct as proposers and Qwen1.5-110B-Chat as aggregators. These are all stronger models than Llama-3.1-8B or Gemma-2-9B. It is common for training on data distilled from stronger models to improve performance.  It would be even more compelling if the trained model could surpass the models corresponding to those data sources, rather than just fitting a better model. However, Table 1 does not compare the performance of MoAA with the most strong two models: WizardLM-8x22B and Qwen1.5-110B-Chat.

**Questions:**

1. What is the performance of WizardLM-8x22B and Qwen1.5-110B-Chat in Table 1?

2. What is the performance of more other reward models in Rewardbench.

3. Why does not compare MoA with MoAA?

---

### Official Review · Reviewer_n9xg · 2024-11-03

**Soundness:** 2
**Presentation:** 2
**Contribution:** 2
**Rating:** 3
**Confidence:** 4

**Summary:**

The paper proposes Mixture of Agents Alignment (MoAA), a scalable and diverse synthetic data recipe that leverages the strengths of various language models to provide high-quality data for model alignment. The method showed improvements over baselines on datasets such as Arena-Hard and AlpacaEval2.

**Strengths:**

1.⁠ ⁠A simple and intuitive approach that leverages multiple open-source models for generating high-quality synthetic data for model alignment.
2.⁠ ⁠The approach demonstrates performance gains over baselines across datasets.
3.⁠ ⁠The paper provides a detailed analysis, offering insights into the impact of each training stage.

**Weaknesses:**

1.⁠ ⁠Limited Novelty: The paper primarily extends existing methodologies, such as the Mixture of Agents (MoA) framework, for generating synthetic data for supervised fine-tuning and preference optimization. Its contributions largely revolve around practical refinements and adaptations of known techniques.
2.⁠ ⁠Weak Baselines: Baselines only consist of open-source instruction-tuned models without comparing against stronger baselines such as [1,2].
3.⁠ ⁠Limited Model Evaluation: Evaluation is limited to Llama 3.1 8B and Gemma 2 9B models. Expanding to a broader range of model sizes and architectures (Llama 3.2 1B and 3B, Qwen 2 0.5B and 1.5B) can provide a good insight into the generalizability of the method.

[1] META-REWARDING LANGUAGE MODELS:
Self-Improving Alignment with LLM-as-a-Meta-Judge, https://arxiv.org/abs/2407.19594v2
[2] MAGPIE: ALIGNMENT DATA SYNTHESIS FROM SCRATCH
BY PROMPTING ALIGNED LLMS WITH NOTHING. https://arxiv.org/abs/2406.08464v1

**Questions:**

See weaknesses.

---

### Official Review · Reviewer_n3xb · 2024-11-04

**Soundness:** 2
**Presentation:** 2
**Contribution:** 2
**Rating:** 5
**Confidence:** 4

**Summary:**

The paper proposes "Mixture of Agents Alignment" (MoAA), a new approach that improves model alignment for large language models through leveraging the collective capabilities of multiple LLMs rather than relying on a single model. Utilizing MoAA, they designed a pipeline for constructing SFT and DPO datasets to improve the target model.

**Strengths:**

1. The performance on AlpacaEval and Arena-Hard is largely improved.
2. MoAA offers a good way to distill the abilities of stronger models into smaller ones.
3. The dataset could be useful to improve open-source models.

**Weaknesses:**

1. The novelty is limited as the MoA was previously proposed and not optimized in this work.
2. There is no ablation on the choice of mixed models.
3. While mixing models can enhance robustness, using this mix as a reward model seems to degrade performance on certain tasks, despite an improved average score driven largely by high safety ratings. However, the number of safety tasks in UC and UF is relatively low, so I'm curious about the difference between using MoA as the reward model versus directly using a strong LLM as the reward model in this context.
4. MoAA’s effectiveness should be validated across additional complex tasks, such as math, reasoning, and coding, to determine its generalizability.

**Questions:**

1. The text format of the model is not consistent in Table 1 and Table 2.
2. It seems the proposed method is not robust to the training data according to Table 2. Could the authors elaborate on why only 5,000 samples from UC were used and how this subset contributed significantly compared to UF alone?
3. Have you evaluated the individual models that comprise MoA?  I wonder if MoAA could outperform the original models.
4. If the target model is the strongest model in MoA, could it be improved?
5. The current MoA models are comprised of different models, can we replace it with the same model of different configs (e.g., system prompt)?
6. The reward model appears less effective than "ArmoRM-Llama3-8B-v0.1." Could incorporating ArmoRM into MoA or replacing the existing model improve alignment?
7. In Table 5, does the best of 5 include the Aggregators?
8. The underperformance of the on-policy setting compared to off-policy is unexpected. Maybe it is due to your hyperparameters.

---

### Official Review · Reviewer_oNHe · 2024-11-04

**Soundness:** 2
**Presentation:** 2
**Contribution:** 2
**Rating:** 5
**Confidence:** 3

**Summary:**

The paper introduces a novel approach called Mixture of Agents Alignment (MoAA) that leverages the collective intelligence of open-source large language models (LLMs) to enhance the model alignment process for building helpful and harmless LLMs. The key contributions include: 1) a two-stage training scheme that uses MoA to generate high-quality synthetic data for supervised fine-tuning (SFT) and preference optimization through direct preference optimization (DPO); 2) extensive evaluations on multiple benchmarks demonstrating significant performance improvements compared to using a single model.

**Strengths:**

(1) The MoAA approach is a novel and promising direction for model alignment, effectively harnessing the diverse capabilities of open-source LLMs through a structured collaboration framework.
(2) The paper demonstrates that the MoA-generated synthetic data leads to substantial performance gains in both SFT and DPO stages, highlighting the value of leveraging collective intelligence.

**Weaknesses:**

(1) While the empirical results are compelling, the paper lacks a more in-depth theoretical analysis of the MoAA approach and its potential limitations.
(2) Lack of comparisons with additional baselines: The evaluations could be strengthened by including comparisons with a wider range of baselines, such as other model ensemble methods or preference optimization techniques.
(3) Potential scalability concerns: The paper does not discuss the computational and memory requirements of the MoA framework, which may be a practical concern when scaling to larger models or datasets.

**Questions:**

(1) Can the authors provide more insights into the design choices for the MoA architecture, such as the optimal number of layers, the specific selection of proposer and aggregator models, and the robustness of the approach to variations in the ensemble composition?
(2) How does the performance of MoAA compare to alternative approaches for generating synthetic data, such as using a single proprietary model or other model ensemble strategies? Can the authors provide additional ablation studies to isolate the contributions of the MoA-based data generation and the DPO stages?
(3) What are the potential limitations or failure modes of the MoAA approach, and how might they be addressed in future work? For example, how would the approach handle cases where the open-source models have significant biases or inconsistencies in their outputs?

---

### Meta-Review · Area_Chair_Baov · 2024-12-21

**Metareview:**

This submission introduces Mixture-of-Agent Alignment, MoAA, an approach that uses multiple LLMs to generate high-quality training data. The motivation is address that cost issue of collecting human-label data as well as its scale, quality, and diversity issues. MoAA addresses these challenges by combining different LLMs' strengths. The authors' results show that it can improve model performance beyond single-model approaches and top closed/ commercial systems.

The reviewers identified the strengths of this work as:
- the proposed approach demonstrates significant performance improvements over baselines.
- the idea of using multiple models to generate higher-quality training data is simple and intuitive.

They also raised concerns on::
- limited technical novelty: several reviewers argue that this work primarily applies the existing MoA framework with minimal new technical contributions;
- incomplete comparisons:  important baselines (e.g., direct training on UltraChat/UltraFeedback) and stronger models are missing from the experiments. in addition, many empirical choices may affect reproducibility.
- generalizability concerns: limited testing on larger models (note that the AC don't consider this as weakness) and other domains such as reasoning related, code or math.

During the rebuttal, the authors managed to convince reviewer n3xb (5) and reviewer UnSv (6) to increase their ratings to 5 and 6, respectively. However, reviewer n3xb (5) still has viewed this work as derived from previous work MoA with further support on its novelty, and reviewer UnSv (6) raised their soundness score and overall score, but kept their  score on technical contribution.

Both reviewer oNHe (5) and reviewer bGhr (5)  did not engage in the rebuttal. reviewer n9xg (3) gave very short review without clarify on the concerns of this submission, and unfortunately this reviewer also did not have engagement to authors' rebuttal. Therefore, this review is disregarded in the recommendation decision.

That being said, this work has final ratings of 6, 5, 5, 5 with major concerns on its technical novelty compared to existing work on MoA and insufficient empirical demonstration during submission. Though many new results and arguments are provided, the reviewers are not convinced in terms of these two issues above. Therefore, this work may significantly benefit from another round of improvement and refinement. All factors considered, this work at its current format is not recommended for acceptance.

**Additional Comments On Reviewer Discussion:**

During the rebuttal, the authors managed to convince reviewer n3xb (5) and reviewer UnSv (6) to increase their ratings to 5 and 6, respectively. However, reviewer n3xb (5) still has viewed this work as derived from previous work MoA with further support on its novelty, and reviewer UnSv (6) raised their soundness score and overall score, but kept their  score on technical contribution.

Both reviewer oNHe (5) and reviewer bGhr (5)  did not engage in the rebuttal. reviewer n9xg (3) gave very short review without clarify on the concerns of this submission, and unfortunately this reviewer also did not have engagement to authors' rebuttal. Therefore, this review is disregarded in the recommendation decision.

---

### Decision · Program_Chairs · 2025-01-22

Reject